# A SEMANTIC LOSS FUNCTION FOR DEEP LEARNING WITH SYMBOLIC KNOWLEDGE

## ABSTRACT

This paper develops a novel methodology for using symbolic knowledge in deep learning. From first principles, we derive a semantic loss function that bridges between neural output vectors and logical constraints. This loss function captures how close the neural network is to satisfying the constraints on its output. An experimental evaluation shows that our semantic loss function effectively guides the learner to achieve (near-)state-of-the-art results on semi-supervised multi-class classification. Moreover, it significantly increases the ability of the neural network to predict structured objects, such as rankings and paths. These discrete concepts are tremendously difficult to learn, and benefit from a tight integration of deep learning and symbolic reasoning methods.

## 1 INTRODUCTION

The widespread success of representation learning raises the question of which AI tasks are amenable to deep learning, which require classical model-based symbolic reasoning, and whether we can benefit from an integration of both. In recent years, significant effort has gone towards various ways of using representation learning to solve tasks that were previously tackled by symbolic methods. Such efforts include neural computers, Turing machines, and differentiable programming (e.g., Weston et al. (2014); Reed & De Freitas (2015); Graves et al. (2016); Riedel et al. (2016)), relational embeddings, deep learning for graph data, and neural theorem proving (e.g., Bordes et al. (2013); Neelakantan et al. (2015); Duvenaud et al. (2015); Niepert et al. (2016)), and many more. Other work has sought to augment deep learning with (symbolic) knowledge (e.g., Hu et al. (2016); Stewart & Ermon (2017); Márquez-Neila et al. (2017); Minervini et al. (2017); Wang et al. (2017)).

This paper considers learning tasks where we have symbolic knowledge connecting the different outputs of a neural network. This knowledge takes the form of a constraint (or sentence) in Boolean logic. It can be as simple as an exactly-one constraint for one-hot output encodings, or as complex as a structured output prediction constraint for intricate combinatorial objects such as rankings, subgraphs, and paths. Our goal is to augment neural networks with the ability to learn how to make predictions subject to these constraints, and use the symbolic knowledge to improve its performance.

Most neuro-symbolic approaches aim to simulate or learn symbolic reasoning in an end-to-end deep neural network, or capture symbolic knowledge in a vector-space embedding. This choice is partly motivated by the need for smooth *differentiable* models; adding symbolic reasoning code (e.g., SAT solvers) to a deep learning pipeline destroys this property. Unfortunately, while making reasoning differentiable, the precise logical meaning of the knowledge is often lost. In this paper, we take a distinctly different approach, and tackle the problem of differentiable but sound logical reasoning from first principles. Starting from a set of intuitive axioms, we derive a differentiable *semantic loss* function that captures how well the outputs of a neural network match a given constraint. This function precisely captures the *meaning* of the constraint, and is independent of its *syntax*.

Next, we show how this semantic loss gives *significant practical improvements* in semi-supervised classification. The semantic loss defined over the exactly-one constraint in this setting permits us to obtain a learning signal from vast amounts of unlabeled data. The key idea is that the semantic loss helps us improve how consistently we are able to classify the unlabeled data. This simple addition to the loss function of standard deep learning architectures yields (near-)state-of-the-art performance in semi-supervised classification on MNIST, FASHION and CIFAR-10 datasets.

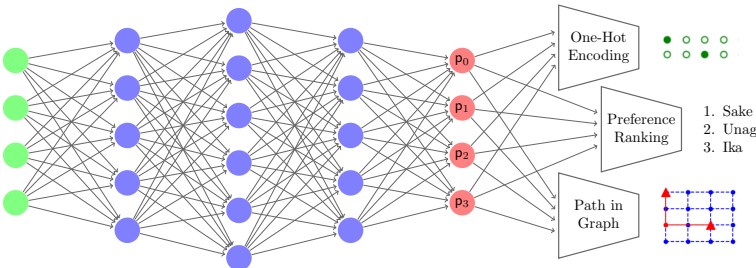

Figure 1: Outputs of a neural network feed into semantic loss functions for constraints representing a one-hot encoding, a total ranking of preferences, and paths in a grid graph.

Our final set of experiments study the benefits of the semantic loss function for complex structured output learning tasks, such as preference learning and path prediction in a graph (Daumé et al., 2009; Chang et al., 2013; Choi et al., 2015; Graves et al., 2016). In these scenarios, the task is two-fold: learn both the structure of the output space, and the actual classification function within that space. By capturing the structure of the output space with logical constraints, and minimizing the semantic loss for this constraint during learning, we are able to learn networks that are much more likely to correctly predict structured objects.

## 2 BACKGROUND AND NOTATION

To formally define semantic loss, we make use of concepts in propositional logic. We write upper-case letters ($X$,$Y$) for Boolean variables and lowercase letters ($x$,$y$) for their instantiation ($X = 0$ or $X = 1$). Sets of variables are written in bold uppercase ($\mathbf{X}$,$\mathbf{Y}$), and their joint instantiation in bold lowercase ($\mathbf{x}$,$\mathbf{y}$). A literal is a variable ($x$) or its negation ($\neg x$). A logical sentence ($\alpha$ or $\beta$) is constructed in the usual way, from variables and logical connectives ($\wedge$, $\vee$, etc.), and is also called a formula or constraint. A state or world $\mathbf{x}$ is an instantiation to all variables $\mathbf{X}$. A state $\mathbf{x}$ satisfies a sentence $\alpha$, denoted $\mathbf{x} \models \alpha$, if the sentence evaluates to be true in that world, as defined in the usual way. A sentence $\alpha$ entails another sentence $\beta$, denoted $\alpha \models \beta$ if all worlds that satisfy $\alpha$ also satisfy $\beta$. A sentence $\alpha$ is logically equivalent to sentence $\beta$, denoted $\alpha \equiv \beta$, if both $\alpha \models \beta$ and $\beta \models \alpha$.

The output row vector of a neural net is denoted $\mathbf{p}$. Each value in $\mathbf{p}$ represents the probability of an output and falls in $[0, 1]$. We use both softmax and sigmoid units for our output activation functions. The notation for states $\mathbf{x}$ is used to refer the an assignment, the logical sentence enforcing the assignment, or the binary vector capturing that same assignment, as these are all equivalent notions.

Figure 1 illustrates the three different concrete output constraints of varying difficulty that are studied in our experiments. First, we examine the exactly-one or *one-hot constraint* capturing the encoding used in multi-class classification. It states that for a set of indicators $\mathbf{X} = \{X_1, \ldots, X_n\}$, one and exactly one of those indicators must be true, with the rest being false. This is enforced through a logical constraint $\alpha$ by conjoining sentences of the form $\neg X_1 \vee \neg X_2$ for all pairs of variables (at most one variable is true), and a single sentence $X_1 \vee \cdots \vee X_n$ (at least one variable is true). Our experiments further examine the *valid simple path constraint*. It states for a given source-destination pair and edge indicators, that the edge indicators which are set to true must form a valid simple path from source to destination. Finally, we explore the *ordering constraint*, which requires that a set of $n^2$ indicator variables represent a total ordering over $n$ variables, effectively encoding a permutation matrix. For a full description of the path and ordering constraints, we refer to Section 5.

## 3 SEMANTIC LOSS

Our goal in this section is to find a semantic loss function that bridges the gap between the continuous world of neural networks, and the symbolic world of propositional logic. We do so by first postulating intuitive high-level properties that we seek in such a function, and that illustrate its desired behavior. A second set of postulates establish a correspondence between constraints and data. Finally, we uniquely define the semantic loss function used throughout this paper.

The semantic loss $L^s(\alpha, p)$ is a function of a sentence $\alpha$ in propositional logic, defined over variables $\mathbf{X} = \{X_1, \ldots, X_n\}$, and a vector of probabilities $p$ for the same variables $\mathbf{X}$. Element $p_i$ denotes the predicted probability of variable $X_i$, and corresponds to a single output of the neural net. For example, the semantic loss between the one-hot constraint from the previous section, and a neural net output vector $p$, is intended to capture how close the prediction $p$ is to having exactly one output set to true (that is, 1), and all others set to false (that is, 0), regardless of which output is correct.

## 3.1 HIGH-LEVEL PROPERTIES

The first axiom says that there is no loss when the logical constraint $\alpha$ is always true (it is a logical tautology), independent of the predicted probabilities $p$.

**Axiom 1** (Truth). The semantic loss of a true sentence is zero: $\forall p, L^s(true, p) = 0$.

Next, when enforcing two constraints on disjoint sets of variables, we want the ability to compute the semantic loss of the two constraints separately, and sum the results for their joint semantic loss.

**Axiom 2** (Additive Independence). Let $\alpha$ be a sentence over $\mathbf{X}$ with probabilities $p$. Let $\beta$ be a sentence over $\mathbf{Y}$ disjoint from $\mathbf{X}$ with probabilities $q$. The semantic loss between sentence $\alpha \wedge \beta$ and the joint probability vector $[p\, q]$ decomposes additively: $L^s(\alpha \wedge \beta, [p\, q]) = L^s(\alpha, p) + L^s(\beta, q)$.

It directly follows from Axioms 1 and 2 that the probabilities of variables that are not used on the constraint do not affect the semantic loss. Proposition 6 in Appendix A formalizes this intuition.

To maintain logical meaning, we postulate that semantic loss is monotone in the order of implication.

**Axiom 3** (Monotonicity). If $\alpha \models \beta$, then the semantic loss $L^s(\alpha, p) \geq L^s(\beta, p)$ for any vector $p$.

Intuitively, as we add stricter requirements to the logical constraint, going from $\beta$ to $\alpha$ and making it harder to satisfy, the semantic loss cannot decrease. For example, when $\beta$ enforces the output of an neural network to encode a subtree of a graph, and we tighten that requirement in $\alpha$ to be a path, the semantic loss cannot decrease. Every path is also a tree and any solution to $\alpha$ is a solution to $\beta$.

A first consequence following the monotonicity axiom is that logically equivalent sentences must incur an identical semantic loss for the same probability vector $p$. Hence, the semantic loss is indeed a semantic property of the logical sentence, and *does not depend on the syntax* of the sentence.

**Proposition 1.** *If $\alpha \equiv \beta$, then the semantic loss $L^s(\alpha, p) = L^s(\beta, p)$ for any vector $p$.*

A second consequence is that semantic loss must be non-negative (see Proposition 5 in Appendix A).

## 3.2 DATA-SENTENCE CORRESPONDENCE

A state $\mathbf{x}$ is equivalently represented as a data vector, as well as a logical constraint that enforces a value for every variable in $\mathbf{X}$. When both the constraint and the predicted vector represent the same state (for example, $X_1 \wedge \neg X_2 \wedge X_3$ vs. $[1\, 0\, 1]$), there should be no semantic loss.

**Axiom 4** (Identity). For any state $\mathbf{x}$, there is zero semantic loss between its representation as a sentence, and its representation as a deterministic vector: $\forall \mathbf{x}, L^s(\mathbf{x}, \mathbf{x}) = 0$.

The axioms above together imply that any vector satisfying the constraint must incur zero loss. For example, when our constraint $\alpha$ requires that the output vector encodes an arbitrary total ranking, and the vector $\mathbf{x}$ correctly represents a single specific total ranking, there is no semantic loss.

**Proposition 2** (Satisfaction). *If $\mathbf{x} \models \alpha$, then the semantic loss $L^s(\alpha, \mathbf{x}) = 0$.*

As a special case, logical literals ($x$ or $\neg x$) constrain a single variable to take on a single value, and thus play a role similar to the labels used in supervised learning. Such constraints require an even tighter correspondence: the semantic loss must act like a classical loss function (i.e., cross entropy).

**Axiom 5** (Label-Literal Correspondence). The semantic loss of a single literal is proportionate to the cross-entropy loss for the equivalent data label: $L^s(x, p) \propto -\log(p)$ and $L^s(\neg x, p) \propto -\log(1-p)$.

Appendix A states Axioms 7 and 8, on the symmetry between values and the symmetry between variables, as well as a Lemma 7 that ties together the multiplicative constants mentioned in Axiom 5. Finally, this allows us to prove the following form of the semantic loss for a state $\mathbf{x}$.

**Lemma 3.** *For state $\mathbf{x}$ and vector $p$, we have $L^s(\mathbf{x}, p) \propto -\sum_{i:\mathbf{x} \models X_i} \log p_i - \sum_{i:\mathbf{x} \models \neg X_i} \log(1 - p_i)$.*

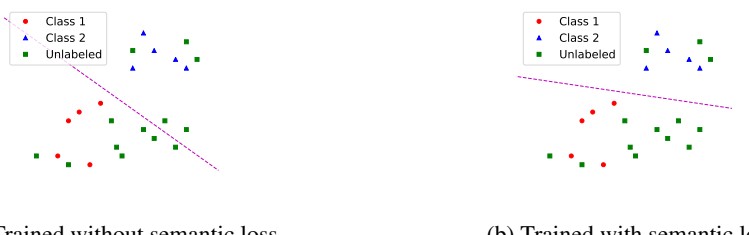

(a) Trained without semantic loss          (b) Trained with semantic loss

Figure 2: Binary classification toy example: a linear classifier without and with semantic loss.

### 3.3 A GENERAL DEFINITION

Lemma 3 falls short as a full definition of semantic loss for arbitrary sentences. One can define additional axioms to pin down $L^s$. For example, the following axiom is highly desirable.

**Axiom 6** (Differentiability). For any fixed $\alpha$, the semantic loss $L^s(\alpha, \mathsf{p})$ is monotone in each probability in $\mathsf{p}$, continuous and differentiable.

Appendix A makes the notion of semantic loss precise by stating one additional axiom. It is based on the observation that the state loss of Lemma 3 is proportionate to a log-probability. In particular, it corresponds to the probability of obtaining state $\mathbf{x}$ after independently sampling each $X_i$ with probability $\mathsf{p}_i$. We have now derived the semantic loss function from first principles as follows.

**Definition 1** (Semantic Loss). Let $\mathsf{p}$ be a vector of probabilities, one for each variable in $\mathbf{X}$, and let $\alpha$ be a sentence over $\mathbf{X}$. The semantic loss between $\alpha$ and $\mathsf{p}$ is

$$L^s(\alpha, \mathsf{p}) \propto -\log \sum_{\mathbf{x} \models \alpha} \prod_{i:\mathbf{x} \models X_i} \mathsf{p}_i \prod_{i:\mathbf{x} \models \neg X_i} (1 - \mathsf{p}_i).$$

**Theorem 4** (Uniqueness). *The semantic loss function in Definition 1 satisfies Axioms 1–9 and is the only function that does so, up to a multiplicative constant.*

Intuitively, the semantic loss is proportionate to a negative logarithm of the probability of generating a state that satisfies the constraint, when sampling values according to $\mathsf{p}$. Hence, it is the self-information (or "surprise") of obtaining an assignment that satisfies the constraint (Jones, 1979).

## 4 SEMI-SUPERVISED CLASSIFICATION

The most straightforward constraint that is ubiquitous in multi-class classification is mutual exclusion over one-hot-encoded outputs. That is, for a given example, exactly one class and therefore exactly one binary indicator must be true. The machine learning community has made great strides in this machine learning task, due to the invention of assorted deep learning representations and their associated regularization terms (Krizhevsky et al., 2012; He et al., 2016). Many of these models take large amounts of fully labeled data for granted, and big data is indispensable for discovering accurate representations (Hastie et al., 2009). To sustain this progress, and alleviate the need for more labeled data, there is a growing interest into utilizing unlabeled data to augment the predictive power of classifiers (Stewart & Ermon, 2017; Bilenko et al., 2004). This section shows why semantic loss naturally qualifies for this task.

**Illustrative Example** To illustrate the benefit of semantic loss in the semi-supervised setting, we begin our discussion with a small toy example. Consider a binary classification task as depicted in Figure 2. Ignoring the unlabeled examples, a simple linear classifier learns to distinguish the two classes by separating the labeled examples in Figure 2a. However, the unlabeled examples are also informative, as they must carry some properties that give them a particular label. This is the crux of semantic loss: a model must confidently assign a consistent class even to unlabeled data. Encouraging the model to do so results in a more accurate decision boundary, as illustrated in Figure 2b. Next, we further explore this idea and apply it to real-world image classification tasks.

Table 1: MNIST. Previously reported test accuracies followed by semantic loss results ($\pm$ stddev)

| Accuracy % with # of used labels | 100 | 1000 | ALL |
|---|---|---|---|
| AtlasRBF (Pitelis et al., 2014) | 91.9 ($\pm$ 0.95) | 96.32 ($\pm$ 0.12) | 98.69 |
| Deep Generative (Kingma et al., 2014) | 96.67($\pm$ 0.14) | 97.60($\pm$ 0.02) | 99.04 |
| Virtual Adversarial (Miyato et al., 2016) | 97.67 | 98.64 | 99.36 |
| Ladder Net (Rasmus et al., 2015) | **98.94** ($\pm$0.37 ) | **99.16** ($\pm$0.08) | **99.43** ($\pm$ 0.02) |
| Baseline: MLP, Gaussian Noise | 78.46 ($\pm$1.94) | 94.26 ($\pm$0.31) | 99.34 ($\pm$0.08) |
| Baseline: Self-Training | 72.55 ($\pm$4.21) | 87.43 ($\pm$3.07 ) | 99.34 ($\pm$0.08) |
| MLP with Semantic Loss (our) | 98.38 ($\pm$0.51) | 98.78 ($\pm$0.17) | 99.36 ($\pm$0.02) |

## 4.1 METHOD

Our proposed method intends to be generally applicable and compatible with any feedforward neural network. The semantic loss is simply another regularization term that can directly be plugged into an existing loss function. More specifically, for some weight $w$, the new overall loss becomes

$$\text{existing loss} + w \cdot \text{semantic loss.}$$

When the constraint over the output space is simple (for example, there is a small number of solutions $\mathbf{x} \models \alpha$), the semantic loss can be directly computed from Definition 1. Concretely, for the exactly-one constraint used in $n$-class classification, the semantic loss reduces to

$$L^s(\text{exactly-one}, \mathsf{p}) \propto -\log \sum_{i=0}^{n-1} \mathsf{p}_i \prod_{j=0,j\neq i}^{n-1} (1 - \mathsf{p}_j),$$

where he values $\mathsf{p}_i$ denote the probability of class $i$ as predicted by the neural net. The semantic loss for the exactly-one constraint is efficient and causes no noticeable overhead in our experiments.

In general, for arbitrary constraints $\alpha$, the semantic loss is not efficient to compute using Definition 1, and more advanced automated reasoning is required. Section 5 discusses this issue in more detail.

## 4.2 EXPERIMENTAL EVALUATION

In this section, we evaluate semantic loss in the semi-supervised setting by comparing it with several competitive models. As most semi-supervised learners build on a supervised learner, changing the underlying model significantly affects the semi-supervised learner's performance. For comparison, we add semantic loss to the same base models used in ladder nets (Rasmus et al., 2015), which currently achieve state-of-the-art results on semi-supervised MNIST and CIFAR-10 (Krizhevsky & Hinton, 2009). Specifically, the MNIST base model is a fully-connected multilayer perceptron (MLP), with layers of size 784-1000-500-250-250-250-10. On CIFAR-10, it is a 10-layer convolutional neural network (CNN) with 3-by-3 padded filters. After every 3 layers, features are subject to a 2-by-2 max-pool layer with strides of 2. Furthermore, we use ReLu (Nair & Hinton, 2010), batch normalization (Ioffe & Szegedy, 2015), and Adam optimization (Kingma & Ba, 2015) with a learning rate of 0.002. We refer to Appendix B and C for a specification of the CNN model and additional details about hyper-parameter tuning.

For all semi-supervised experiments, we use the standard 10,000 held-out test examples provided in the original datasets and randomly pick 10,000 from the standard 60,000 training examples (50,000 for CIFAR-10) as validation set. For values of $N$ that depend on the experiment, we retain $N$ randomly chosen labeled examples from the training set, and remove labels from the rest. We balance classes in the labeled samples to ensure no particular class is over-represented. Images are preprocessed for standardization and Gaussian noise (standard deviation 0.3) is added to every pixel.

**MNIST** The permutation invariant MNIST classification task is commonly used as a test-bed for general semi-supervised learning algorithms. This setting does not use any prior information about the spatial arrangement of the input pixels. Therefore, it excludes many data augmentation techniques that involve geometric distortion of images, as well as convolutional neural networks.

When evaluating on MNIST, we run experiments for 10 epochs, with a batch size of 10 labeled and 10 unlabeled examples. Experiments are repeated 10 times with different random seeds. Table 1

Table 2: FASHION. Test accuracy comparison between MLP with semantic loss and ladder nets.

| Accuracy % with # of used labels | 100 | 500 | 1000 | ALL |
|---|---|---|---|---|
| Ladder Net (Rasmus et al., 2015) | 81.46 ($\pm0.64$ ) | 85.18 ($\pm0.27$) | 86.48 ($\pm$ 0.15) | **90.46** |
| Baseline: MLP, Gaussian Noise | 69.45 ($\pm2.03$) | 78.12 ($\pm1.41$) | 80.94 ($\pm0.84$) | 89.87 |
| MLP with Semantic Loss (our) | **86.74** ($\pm0.71$) | **89.49** ($\pm0.24$) | **89.67** ($\pm0.09$) | 89.81 |

Table 3: CIFAR. Test accuracy comparison between CNN with semantic loss and ladder nets.

| Accuracy % with # of used labels | 4000 | ALL |
|---|---|---|
| CNN Baseline in Ladder Net | 76.67 ($\pm$ 0.61) | 90.73 |
| Ladder Net (Rasmus et al., 2015) | 79.60 ($\pm0.47$) | |
| Baseline: CNN, Whitening, Cropping | 77.13 | **90.96** |
| CNN with Semantic Loss (our) | **81.79** | 90.92 |

compares semantic loss to two baselines and state-of-the-art results from the literature. The first baseline is a purely supervised MLP, which makes no use of unlabeled data. The second is the classic self-training method for semi-supervised learning, which operates as follows. After every 1000 iterations, the unlabeled examples that are predicted by the MLP to have more than $95\%$ probability of belonging to a single class, are assigned a psuedo-label and become labeled data.

When given 100 labeled examples ($N = 100$), MLP with semantic loss gains around $20\%$ improvement over the purely supervised baseline. The improvement is even larger ($25\%$) compared to self-training. Considering *the only change is an additional loss term*, this result is very encouraging. Compared to the state of the art, ladder nets slightly outperform semantic loss by $0.5\%$ accuracy. This difference may be an artifact of the excessive tuning of architectures, hyper-parameters and learning rates that the MNIST dataset has been subject to. In the coming experiments, we extend our work to more challenging datasets, in order to provide a clearer comparison with ladder nets.

First, we want to share a few more thoughts on how semantic loss works. A classical softmax layer interprets its output as representing a categorical distribution. Hence, by normalizing its outputs, softmax enforces the same mutual exclusion constraint enforced in our semantic loss function. However, there does not exist a natural way to extend softmax loss to unlabeled samples. In contrast, semantic loss does provide a learning signal on unlabeled samples, by forcing the underlying classifier to make an decision and construct a confident hypothesis for all data. However, for the fully supervised case ($N = $ all), the semantic loss does not significantly affect accuracy. Because the MLP has enough capacity to almost perfectly fit the training data, where the constraint is always satisfied, the semantic loss is almost always zero. This is a direct consequence of Proposition 2.

**FASHION** The FASHION (Xiao et al., 2017) dataset consists of Zalando's article images, aiming to serve as a more challenging drop-in replacement for MNIST. Arguably, it has not been overused and requires more advanced techniques to achieve good performance. As in the previous experiment, we run our method for 10 epochs, whereas ladder nets need 100 epochs to converge. Again, experiments are repeated 10 times and Table 2 reports the classification accuracy and its standard deviation (except for $N = $ all where it is close to $0$ and omitted for space).

Experiments show that utilizing semantic loss results in a very large $17\%$ improvement over the baseline when only 100 labels are provided. Moreover, our method compares favorably to ladder nets, except when the setting degrades to be fully supervised. Note that our method already nearly reaches its maximum accuracy with 500 labeled examples, which is only $1\%$ of the training dataset.

**CIFAR-10** To show the general applicability of semantic loss, we evaluate it on CIFAR-10. This dataset consisting of 32-by-32 RGB images in 10 classes. A simple MLP would not have enough representation power to capture the huge variance across objects within the same class. To cope with this spike in difficulty, we switch our underlying model to a 10-layer CNN as described earlier. We use a batch size of 100 samples of which half are unlabeled. Experiments are run for 100 epochs. However, due to our limited computational resources, we report on a single trial. Note that we make slight modifications to the underlying model used in ladder nets to reproduce similar baseline performance. Please refer to Appendix B for the details of this experimental setup.

As shown in Table 3, our method compares favorably to ladder nets. However, due to the slight difference in performance between the supervised base models, a direct comparison would be methodologically flawed. Instead, we compare the net improvements over baselines. In terms of this measure, our method scores a gain of $4.66\%$ whereas ladder nets gain $2.93\%$.

## 4.3 DISCUSSION

Overall, the experiments so far have demonstrated the competitiveness and general applicability of our proposed method on semi-supervised learning tasks. It surpassed the previous state of the art (i.e. ladder nets ) on FASHION and CIFAR-10, while being close on MNIST. Considering the simplicity of our method, such results are encouraging. Indeed, a key advantage of semantic loss is that it only requires a simple additional loss term. Without changing the network architecture itself, we incur almost no computational overhead. Conversely, this property makes our method sensitive to the underlying model's performance. Without the underlying predictive power of a strong supervised learning model, we do not expect to see the same benefits we observe here. Recently, we became aware that Miyato et al. (2016) extended their work to CIFAR-10 and achieved state-of-the-art results (Miyato et al., 2017), surpassing our performance by $5\%$. In future work, we plan to investigate whether applying semantic loss on their architecture would yield an even stronger performance.

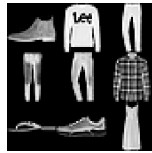 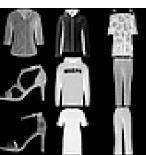 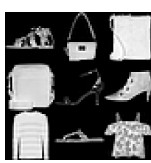 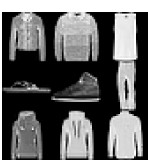

(a) Confidently Correct    (b) Unconfidently Correct  (c) Unconfidently Incorrect  (d) Confidently Incorrect

Figure 3: FASHION pictures grouped by how confidently the supervised base model classifies them correctly. With semantic loss, the final semi-supervised model predicts all correctly and confidently.

Figure 3 illustrates the effect of semantic loss on FASHION pictures whose correct label was hidden from the learner. Pictures 3a and 3b are correctly classified by the supervised base model, and on the first set it is confident about this prediction ($p_i > 0.8$). The semantic loss rarely diverts the model from these initially correct labels. However, it bootstraps these unlabeled examples to achieve higher confidence in the learned concepts. With this additional learning signal, the model changes its beliefs about Pictures 3c, which it was previously uncertain about. Finally, even on confidently misclassified Pictures 3d, the semantic loss is able to fully correct the mistakes of the base model.

## 5 LEARNING WITH COMPLEX CONSTRAINTS

While much of current machine learning research is focused on problems such as multi-class classification, there remain a multitude of difficult problems involving highly constrained output domains. As mentioned in the previous section, semantic loss has little effect on the fully-supervised exactly-one classification problem. This leads us to seek out more difficult problems to illustrate that semantic loss can also be highly informative in the supervised case, provided the output domain is a sufficiently complex space. Because semantic loss is defined by a Boolean formula, it can be used on any output domain that can be fully described in this manner. Here, we develop a framework for tractable semantic loss on highly complex constraints, and evaluate it on some difficult examples.

## 5.1 A TRACTABLE SEMANTIC LOSS

Our goal here is to develop a method for computing both the semantic loss and its gradient in a tractable manner. Examining Definition 1 of semantic loss, we see that the right-hand side is a well-known automated reasoning task called weighted model counting (WMC) (Chavira & Darwiche, 2008; Sang et al., 2005). A key property of WMC is that its partial derivatives can be computed in terms of other, slightly modified WMCs. Furthermore, we know of circuit languages that compute WMCs, and that are amenable to backpropagation (Darwiche, 2003). We use the language and circuit compilation techniques described in Darwiche (2011) to build a Boolean circuit representing

Table 4: Grid shortest path test results: coherent, incoherent and constraint accuracy.

| Test accuracy % | Coherent | Incoherent | Constraint |
|---|---|---|---|
| 5-layer MLP | 5.62 | **85.91** | 6.99 |
| With semantic loss (our) | **28.51** | 83.14 | **69.89** |

Table 5: Preference prediction test results: coherent, incoherent and constraint accuracy.

| Test accuracy % | Coherent | Incoherent | Constraint |
|---|---|---|---|
| 3-layer MLP | 1.01 | **75.78** | 2.72 |
| With semantic loss (our) | **13.59** | 72.43 | **55.28** |

our semantic loss. We refer to the literature for details of this compilation approach. Due to certain properties of this circuit form, we can use it to compute both the values and the gradients of the semantic loss in time linear in the size of the circuit (Darwiche & Marquis, 2002). Once we have constructed this function, we can add it to our standard loss function as described in Section 4.1.

## 5.2 EXPERIMENTAL EVALUATION

Our ambition when evaluating semantic loss' performance on complex constraints is not to achieve state-of-the-art performance on any particular problem, but rather to highlight its effect. To this end, we evaluate our method on problems with a difficult output space, where the model could no longer be fit directly from data, and purposefully use simple MLPs for evaluation. The details of hyper-parameter tuning are again given in Appendix C.

### 5.2.1 GRIDS

We begin with a classic algorithmic problem, finding the shortest path in a graph. Specifically, we use a 4-by-4 grid $G = (V, E)$ with uniform edge weights. We randomly remove edges for each example to increase difficulty. Formally, our input is a binary vector of length $|V| + |E|$, with the first $|V|$ variables indicating sources and destinations, and the next $|E|$ which edges are removed. Similarly, each label is a binary vector of length $|E|$ indicating which edges are in the shortest path. Finally, we require through our constraint $\alpha$ that the output form a valid simple path between the desired source and destination. To compile this constraint, we use the method of Nishino et al. (2017) to encode pairwise simple paths, and logically merge them to enforce the correct source and destination. For more details on the constraint and data generation process, see Appendix D.

To evaluate, we use our generated dataset of 1600 examples, with a 60/20/20 train/validation/test split. Table 4 compares test accuracy between a 5-layer MLP baseline, and the same model augmented with semantic loss. We report three different accuracies that illustrate the effect of semantic loss: "Coherent" indicates the percentage of examples for which the classifier gets the entire configuration right, while "Incoherent" measures the percentage of individually correct binary labels, which as a whole may not constitute a valid path at all. Finally, "Constraint" describes the percentage of predictions given by the model that satisfy the constraint associated with the problem. In the case of incoherent accuracy, semantic loss has little effect, and in fact slightly reduces the accuracy as it combats the standard sigmoid cross entropy. In regard to coherent accuracy, however, the semantic loss has a very large effect in guiding the network to jointly learn true paths, rather than optimizing each binary output individually. We further see this by observing the large increase in the percentage of predictions which really are paths between the desired nodes in the graph.

### 5.2.2 PREFERENCE LEARNING

The next problem we examine is that of predicting a complete order of preferences. That is, for a given set of user features, we would like to predict how the user would rank their preference over a fixed set of items. We encode a preference ordering over $n$ items as a flattened binary matrix $\{X_{ij}\}$, where for each $i, j \in \{1, \ldots, n\}$, $X_{ij}$ denotes that item $i$ is at position $j$ (Choi et al., 2015). Clearly, not all configurations of outputs correspond to a valid ordering.

For data, we use preference rankings over 10 types of sushi for 5000 individuals, taken from PREFLIB (Mattei & Walsh, 2013). We take the ordering over 6 types of sushi as input features to predict the ordering over the remaining 4 types, with splits identical to those in Shen et al. (2017). We again split the data 60/20/20 into train/test/split, and employ a 3 layer MLP as our baseline. Table 5 compares the baseline to the same MLP augmented with semantic loss for valid total orderings. Again, we see that semantic loss has a marginal effect on incoherent accuracy, but massively improves the network's ability to predict valid, correct orderings. Remarkably, without semantic loss, the network is only able to output a valid ordering on $0.01\%$ of the test examples.

## 6 RELATED WORK

Incorporating symbolic background knowledge into machine learning is a long-standing challenge (Srinivasan et al., 1995). It has received considerable attention for structured prediction in natural language processing, in both supervised and semi-supervised settings. For example, *constrained conditional models* extend linear models with constraints that are enforced through integer linear programming (Chang et al., 2008; 2013). Constraints have also been studied in the context of probabilistic graphical models (Mateescu & Dechter, 2008; Ganchev et al., 2010). Kisa et al. (2014) utilize a circuit language called the *probabilistic sentential decision diagram* to induce distributions over arbitrary logical formulas. They learn generative models that satisfy preference and path constraints (Choi et al., 2015; 2016), which we both study in a discriminative setting.

Various deep learning techniques have been proposed to enforce either arithmetic constraints (Pathak et al., 2015; Márquez-Neila et al., 2017) or logical constraints (Rocktäschel et al., 2015; Hu et al., 2016; Demeester et al., 2016; Stewart & Ermon, 2017; Minervini et al., 2017; Diligenti et al., 2017; Donadello et al., 2017) on the output of a neural network. The common approach is to reduce logical constraints into differentiable arithmetic objectives by replacing logical operators with their fuzzy t-norms and logical implications with simple inequalities. A downside of this fuzzy relaxation is that the logical sentences lose their precise meaning. The learning objective becomes a function of the syntax rather than the semantics. Moreover, these relaxations are often only applied to Horn clauses. One alternative is to encode the logic into a factor graph and perform loopy belief propagation to compute a loss function (Naradowsky & Riedel, 2017), which is known to have issues in the presence of complex logical constraints (Smith & Gogate, 2014).

Several specialized techniques have been proposed to exploit the rich structure of real world labels. Deng et al. (2014) propose hierarchy and exclusion graphs that allow a flexible joint modeling of hierarchical categories. It is a method invented to address examples whose labels are not provided at the most specific level. Finally, the objective of semantic loss to increase the confidence of predictions on unlabeled data is in common with information-theoretic approaches to semi-supervised learning (Grandvalet & Bengio, 2005; Erkan & Altun, 2010), and approaches that increase robustness to output perturbation (Miyato et al., 2016). A key difference between semantic loss and these information-theoretic losses is that semantic loss generalizes to arbitrary output constraints.

## 7 CONCLUSIONS

Both reasoning and semi-supervised learning are often identified as key challenges for deep learning going forward. In this paper, we developed a principled way of combining automated reasoning for propositional logic with existing deep learning architectures. Moreover, we showed that our semantic loss function provides significant benefits during semi-supervised classification, as well as deep structured prediction for highly complex output spaces.

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

## A  AXIOMATIZATION OF SEMANTIC LOSS: DETAILS

This appendix provides further details on our axiomatization of the semantic loss.

**Proposition 5** (Non-Negativity). *Semantic loss is non-negative.*

*Proof.* Because $\alpha \models true$ for all $\alpha$, the monotonicity axiom implies that $\forall p, L^s(\alpha, p) \geq L^s(true, p)$. By the truth axiom, $L^s(true, p) = 0$, and therefore $L^s(\alpha, p) \geq 0$ for all choices of $\alpha$ and $p$. □

**Proposition 6** (Locality). *Let $\alpha$ be a sentence over $\mathbf{X}$ with probabilities $p$. For any $\mathbf{Y}$ disjoint from $\mathbf{X}$ with probabilities $q$, the semantic loss $L^s(\alpha, [p\,q]) = L^s(\alpha, p)$.*

*Proof.* Follows from the additive independence and truth axioms. Set $\beta = true$ in the additive independence axiom, and observe that this sets $L^s(\beta, q) = 0$ because of the truth axiom. □

*Proof of Proposition 2.* The monotonicity axiom specializes to say that if $\mathbf{x} \models \alpha$, we have that $\forall p, L^s(\mathbf{x}, p) \geq L^s(\alpha, p)$. By choosing $p$ to be $\mathbf{x}$, this implies $L^s(\mathbf{x}, \mathbf{x}) \geq L^s(\alpha, \mathbf{x})$. From the identity axiom, $L^s(\mathbf{x}, \mathbf{x}) = 0$, and therefore $0 \geq L^s(\alpha, \mathbf{x})$. Proposition 5 bounds the loss from below as $L^s(\alpha, \mathbf{x}) \geq 0$. □

**Axiom 7** (Value Symmetry). For all $\mathsf{p}$ and $\alpha$, we have that $\mathrm{L}^s(\alpha, \mathsf{p}) = \mathrm{L}^s(\bar{\alpha}, 1-\mathsf{p})$ where $\bar{\alpha}$ replaces every variable in $\alpha$ by its negation.

**Axiom 8** (Variable Symmetry). Let $\alpha$ be a sentence over $\mathbf{X}$ with probabilities $\mathsf{p}$. Let $\pi$ be a permutation of the variables $\mathbf{X}$, let $\pi(\alpha)$ be the sentence obtained by replacing variables $x$ by $\pi(x)$, and let $\pi(\mathsf{p})$ be the corresponding permuted vector of probabilities. Then, $\mathrm{L}^s(\alpha, \mathsf{p}) = \mathrm{L}^s(\pi(\alpha), \pi(\mathsf{p}))$.

The value and variable symmetry axioms together imply the equality of the multiplicative constants in the label-literal duality axiom for all literals.

**Lemma 7.** *There exists a single constant $K$ such that $\mathrm{L}^s(X, p) = -K \log(p)$ and $\mathrm{L}^s(\neg X, p) = -K \log(1 - p)$ for any literal $x$.*

*Proof.* Value symmetry implies that $\mathrm{L}^s(X_i, \mathsf{p}) = \mathrm{L}^s(\neg X_i, 1 - \mathsf{p})$. Using label-literal correspondence, this implies $K_1 \log(p_i) = K_2 \log(1 - (1 - p_i))$ for the multiplicative constants $K_1$ and $K_2$ that are left unspecified by that axiom. This implies that the constants are identical. A similar argument based on variable symmetry proves equality between the multiplicative constants for different $i$. $\quad\square$

*Proof of Lemma 3.* A state $\mathbf{x}$ is a conjunction of independent literals, and therefore subject to the additive independence axiom. Each literal's loss in this sum is defined by Lemma 7. $\quad\square$

The following and final axiom requires that the semantic loss is proportionate to the logarithm of a function that is additive for mutually exclusive sentences.

**Axiom 9** (Exponential Additivity). Let $\alpha$ and $\beta$ be mutually exclusive sentences (i.e., $\alpha \wedge \beta$ is unsatisfiable), and let $f^s(K, \alpha, \mathsf{p}) = K^{-\mathrm{L}^s(\alpha, \mathsf{p})}$. Then, there exists a positive constant $K$ such that $f^s(K, \alpha \vee \beta, \mathsf{p}) = f^s(K, \alpha, \mathsf{p}) + f^s(K, \beta, \mathsf{p})$.

*Proof of Theorem 4.* The truth axiom states that $\forall \mathsf{p}, f^s(K, true, \mathsf{p}) = 1$ for all positive constants $K$. This is the first Kolmogorov axiom of probability. The second Kolmogorov axiom for $f^s(K, ., \mathsf{p})$ follows from the additive independence axiom of semantic loss. The third Kolmogorov axiom (for the finite discrete case) is given by the exponential additivity axiom of semantic loss. Hence, $f^s(K, ., \mathsf{p})$ is a probability distribution for some choice of $K$, which implies the definition up to a multiplicative constant. $\quad\square$

## B  Specification of the Convolutional Neural Network Model

Table 6 shows the slight architectural difference between the CNN used in ladder nets and ours. The major difference lies in the choice of ReLu. Note we add standard padded cropping to preprocess images and an additional fully connected layer at the end of the model, neither is used in ladder nets. We only make those slight modification so that the baseline performance reported by Rasmus et al. (2015) can be reproduced.

## C  Hyper-parameter Tuning Details

Validation sets are used for tuning the weight associated with semantic loss, the only hyper-parameter that causes noticeable difference in performance for our method. For our semi-supervised classification experiments, we perform a grid search over $\{0.001, 0.005, 0.01, 0.05, 0.1\}$ to find the optimal value. Empirically, $0.005$ always gives the best or nearly the best results and we report its results on all experiments.

For the FASHION dataset specifically, because MNIST and FASHION share the same image size and structure, methods developed in MNIST should be able to directly perform on FASHION without heavy modifications. Because of this, we use the same hyper-parameters when evaluating our method. However, for the sake of fairness, we subject ladder nets to a small-scale parameter tuning in case its performance is more volatile.

For the grids experiment, the only hyper parameter that needed to be tuned was again the weight given to semantic loss, which after trying $\{0.001, 0.005, 0.01, 0.05, 0.1, 0.5, 1\}$ was selected to be

Table 6: Specifications of CNNs in Ladder Net and our proposed method.

| CNN in Ladder Net | CNN in this paper |
|---|---|
| Input 32×32 RGB image | |
| | Resizing to $36 \times 36$ with padding |
| | Cropping Back |
| Whitening | |
| Contrast Normalization | |
| Gaussian Noise with std. of 0.3 | |
| 3×3 conv. 96 BN LeakyReLU | 3×3 conv. 96 BN ReLU |
| 3×3 conv. 96 BN LeakyReLU | 3×3 conv. 96 BN ReLU |
| 3×3 conv. 96 BN LeakyReLU | 3×3 conv. 96 BN ReLU |
| 2×2 max-pooling stride 2 BN | |
| 3×3 conv. 192 BN LeakyReLU | 3×3 conv. 192 BN ReLU |
| 3×3 conv. 192 BN LeakyReLU | 3×3 conv. 192 BN ReLU |
| 3×3 conv. 192 BN LeakyReLU | 3×3 conv. 192 BN ReLU |
| 2×2 max-pooling stride 2 BN | |
| 3×3 conv. 192 BN LeakyReLU | 3×3 conv. 192 BN ReLU |
| 1×1 conv. 192 BN LeakyReLU | 3×3 conv. 192 BN ReLU |
| 1×1 conv. 10 BN LeakyReLU | 1×1 conv. 10 BN ReLu |
| global meanpool BN | |
| | fully connected BN |
| 10-way softmax | |

0.5 based on validation results. For the preference learning experiment, we initially chose the semantic loss weight from $\{0.001, 0.005, 0.01, 0.05, 0.1, 0.5, 1\}$ to be 0.1 based on validation, and then further tuned the weight to 0.25.

## D  SPECIFICATION OF COMPLEX CONSTRAINT MODELS

**Grids**   To compile our grid constraint, we first use Nishino et al. (2017) to generate a constraint for each source destination pair. Then, we conjoin each of these with indicators specifying which source and destination pair must be used, and finally we disjoin all of these together to form our constraint.

To generate the data, we begin by randomly removing one third of edges. We then filter out connected components with fewer than 5 nodes to reduce degenerate cases, and proceed with randomly selecting pairs of point to create data points.

The predictive model we employ as our baseline is a 5 layer MLP with 50 hidden sigmoid units per layer. It is trained using Adam Optimizer, with full data batches (Kingma & Ba, 2015). Early stopping with respect to validation loss is used as a regularizer.

**Preference Learning**   We split each user's ordering into their ordering over sushis 1,2,3,5,7,8, which we use as the features, and their ordering over 4,6,9,10 which are the labels we predict. The constraint is compiled directly from logic, as this can be done in a straightforward manner for an n-item ordering.

The predictive model we use here is a 3 layer MLP with 25 hidden sigmoid units per layer. It is trained using Adam Optimizer with full data batches (Kingma & Ba, 2015). Early stopping with respect to validation loss is used as a regularizer.

