# OpenReview forum: "A Semantic Loss Function for Deep Learning with Symbolic Knowledge"
_ICLR.cc/2018/Conference — Reject_

### Official Review · AnonReviewer3 · 2017-11-26
**Deep learning with symbolic information**

**Rating:** 7
**Confidence:** 3

**Review:**

SUMMARY

The paper proposes a new form of regularization utilizing logical constraints. The semantic loss function is built on the exploitation of symbolic knowledge extracted from data and connecting the logical constraints to the outputs of a neural network. The use of Boolean logic as a constraint provides a secondary regularization term to prevent over-fitting and improve predictions. The benefit of using the function is found primarily with semi-supervised tasks where data is partially unlabelled. The logical constraints provided by the semantic loss function allow for improved classification of unlabeled data.
Output constraints for the semantic loss function are represented with one-hot encoding, prefer- ence rankings, and paths in a grid. These three different output constraints are designed to explore different learning purposes. The semantic function was tested on both semi-supervised classifica- tion tasks as well as structure learning. The paper primarily focuses on the one-hot encoding constraint as it is viewed as a capable technique for multi-class classification.

POSITIVES

In terms of structure, the paper was written very well. Sufficient background information was con- veyed which helped in understanding the proposed semantic loss function. A thorough breakdown is also carried out on the semantic loss function itself by explaining its axioms which help explain how the outputs of a neural network match a given constraint.
As a scientific contribution, I would say results from the experiments were able to justify the proposal of the semantic loss function. The function was able to perform better than most other implementations for semi-supervised learning tasks, and the function was tested on multiple datasets. The paper also made use of testing the function against other notable machine learning approaches, and in most cases the function performed better, but this usually was confined to semi-supervised learning tasks. During supervised learning tasks the function did not perform markedly better than older implementations. Given that, the semantic loss function did prove to be a seemingly simple approach to improving semi-supervised classification tasks.
• The background section covers the knowledge required in understanding the semantic loss function. The paper also clearly explains the meaning for some of the notation used in the definitions.
• Experiments which clearly show the benefit of using the semantic loss function. Multiple experiment types were done as well which showed evidence of the broad applicability of the function.
• In depth description of the definitions, axioms, and propositions of the semantic loss function.
• A large number of experiments exploring the usefulness of the function for multiple learning tasks, and on multiple datasets.

NEGATIVES

I was not clear if the logical constraints are to be instantiated before learning, i.e. they are defined by hand prior to being implemented in the neural network. This is a pretty important question and drastically changes the nature of the learning process. Beyond that complaint, the paper did not suffer from any critical issues. There were some issues with spelling, and the section titled ’Algorithm’ fails to clearly define a complete algorithm using the semantic loss function. It would have helped to have two algorithms. One defining the pipeline for the semantic loss function, and another showing the implementation of the function in a machine learning framework. The semantic loss function found success only in cases were the learning task was semi-supervised, and not in cases of total supervised learning. This is not a true negative, but an observation on the effectiveness of the function.

- A few typos in the paper.
- The axioms for the semantic loss function where defined but there seemed to be a lack of a clear algorithm provided showing the pipeline implementation of the semantic loss function.
- While the semantic loss function does improve learning performance in most cases, the im- provements are confined to semi-supervised learning tasks, and with the MNIST dataset another methodology, Ladder Nets, was able to outperform the semantic loss function.

RELATED WORK

The paper proposed that logic constraints applied to the output of neural networks have the capacity to improve semi-supervised classification tasks as well as finding the shortest path. In the introduction, the paper lists Zhiting Hu et al. paper titled Harnessing Deep Neural Networks with Logic Rules as an example of a similar approach. Hu et al. paper utilized logic constraints in conjunction with neural nets as well. A key difference was that Hu et al. applied their network architecture to supervised classification tasks. Since the performance of the current papers semantic loss function with supervised tasks did not improve upon other methods, it may benefit to utilize the research by Hu et al. as a means of direct comparison for supervised learning tasks, and possibly incorporate their methods with the semantic loss function in order to improve upon supervised learning tasks.

CONCLUSION

Given the success of the semantic loss function with semi-supervised tasks, I would accept this paper. The semantic loss was able to improve learning with respect to the tested datasets, and the paper clearly described the properties of the functions. The paper would benefit by including a more concrete algorithm describing the flow of data through a given neural net to the semantic loss function, as well as the process by which the semantic loss function constrains the data based on propositional logic, but in general this complaint is more nit picking. The semantic loss function and the experiments which tested the function showed clearly that there is a benefit to this research and there are areas for it to improve.

---

> ### Author Response · Authors · 2017-12-31
> **Response to AnonReviewer3**
>
> Thank you for your valuable feedback.
>
> There is an important point of misunderstanding that we would like to clarify. Our supervised learning experiments actually do show a significant improvement over the underlying baseline MLP. The neural network’s ability to predict shortest paths improves from 5% to 28% accuracy. Its ability to predict total rankings improves from 1% to 13% accuracy. In the revised paper we have attempted to better highlight these results. The column for “coherent loss” is the one to look at. The “incoherent loss” is not affected much by semantic loss, and represents the accuracy of predicting individual edges in the graph, not paths. For this easier problem, the output constraint is irrelevant. We have also added a column highlighting the “constraint accuracy”, for the fraction of test outputs that satisfy the constraint. That accuracy improves very significantly with semantic loss, from 7% to 70% and from 3% to 55%, showing that semantic loss does help the deep net in the supervised setting to learn the concept expressed by the logical constraint.
>
> In our experiments, the constraints were always provided by the user before learning (they are inherent to the task). Trying to learn the constraints directly from data is an interesting idea for future work. Although learning theory tells us this can be quite challenging.
>
> We agree that a comparison with related work was missing from the initial submission. We have now added an extensive discussion of related work and how it is conceptually different from semantic loss in the revised paper we posted (see Section 6). We have also included 17 new references.
>
> Specifically for the Hu et al. paper you mention, the key difference with semantic loss is that Hu et al. use fuzzy logic. This has two implications. First, the fuzzy loss function is very sensitive to the syntax of the logical constraint, whereas our loss only depends on the semantics. Second, the logical constraints supported by these fuzzy alternatives are much more simple than the ones we consider. For example, for the Grids experiment in our paper, the constraint is very complex (it doesn’t even have a compact CNF form), and needs to be represented as a logical circuit (see Nishino et al. 2017). We are not aware of a reasonable fuzzy logic encoding. In contrast, other work that uses fuzzy logic to encode constraints works with very simple logical sentences (usually simple implications (X => Y) or Horn clauses). We have also attempted to compare experimentally on the benchmarks of Hu et al. Initial experiments suggested that semantic loss outperforms the loss used by Hu et al. on their evaluation tasks. However, because we found it difficult to exactly reproduce the initialization of Hu et al. and were not able to perform enough tuning, we prefer to not report on that experiment at this time.
>
> In the revised paper, we have also tried to address your comments about the algorithm section, spelling mistakes, etc.

---

### Official Review · AnonReviewer2 · 2017-11-28
**The paper proposes a new loss function that penalizes semantic features of the data, and shows some experiments; overall the writing is good and the ideas are nice, even though the contribution is relatively small.**

**Rating:** 5
**Confidence:** 3

**Review:**

The authors propose a new loss function that is directed to take into account Boolean constraints involving the variables of a classification problem. This is a nice idea, and certainly relevant. The authors clearly describe their problem, and overall the paper is well presented. The contributions are a loss function derived from a set of axioms, and experiments indicating that this loss function captures some valuable elements of the input. This is a valid contribution, and the paper certainly has some significant strengths.

Concerning the loss function, I find the whole derivation a bit distracting and unnecessary. Here we have some axioms, that are not simple when taken together, and that collectively imply a loss function that makes intuitive sense by itself. Well, why not just open the paper with Definition 1, and try to justify this definition on the basis of its properties. The discussion of axioms is just something that will create debate over questionable assumptions. Also it is frustrating to see some axioms in the main text, and some axioms in the appendix (why this division?).

After presenting the loss function, the authors consider some applications. They are nicely presented; overall the gains are promising but not that great when compared to the state of the art --- they suggest that the proposed semantic loss makes sense. However I find that the proposal is still in search of a "killer app". Overall, I find that the whole proposal seems a bit premature and in need of more work on applications (the work on axiomatics is fine as long as it has something to add).

Concerning the text, a few questions/suggestions:
- Before Lemma 3, "this allows..." is the "this including the other axioms in the appendix?
- In Section 4, line 3: I suppose that the constraint is just creating a problem with a class containing several labels, not really a multi-label classification problem (?).
- The beginning of Section 4.1 is not very clear. By reading it, I feel that the best way to handle the unlabeled data would be to add a direct penalty term forcing the unlabeled points to receive a label. Is this fair?
- Page 6: "a mor methodological"... should it be "a more methodical"?
- There are problems with capitalization in the references. Also some references miss page numbers and some do not even indicate what they are (journal papers, conference papers, arxiv, etc).

---

> ### Author Response · Authors · 2017-12-31
> **Response to AnonReviewer2**
>
> Thank you for your valuable feedback.
>
> About searching for a “killer-app”: our results on semi-supervised learning are either very close to the state-of-art (0.5% lower) or exceed it by a significant margin (see Tables 2 and 3). Moreover, our approach is a lot simpler than previous ones: it does not require any special-purpose architecture or customized learner, has only one parameter to tune, and is easy to apply to any network. The fact that you can just add our loss function to an existing architecture and get results as good as state-of-the-art special-purpose techniques makes semi-supervised learning a “killer-app” for semantic loss. We believe semantic loss has the potential of becoming the standard initial-trial method for semi-supervised learning, because of its simplicity and effectiveness. To be fair, our experiments on Path and Preference learning are indeed more academic, but also show promise on highly symbolic problems that are very hard to solve with deep learning.
>
> Whether to derive the semantic loss from axioms or simply state it is largely a matter of taste. The axiomatic approach is certainly more common in basic mathematics and formal logic. We hope to convince you that our axioms implying the definition has the following advantage. When someone proposes an alternative loss function to enforce logical constraints, they will have to violate at least one of our axioms, which will make it clear what the difference is in their assumptions. For example, Axiom 3 will be violated by most loss functions based on fuzzy logic (see related work). Generally, we find this to be a proper scientific way of defining new concepts (versus stating some arbitrary function).
> We agree that it is inconvenient that the axioms are split over main text and appendix. We felt obliged to do so in order to stay within the recommended page limit for ICLR. If the reviewers recommend to add all axioms to the main text, we are happy to make that change.
>
> Addressing your specific questions:
> - Lemma 3 follows from Axioms 1-5 in the main text and Axioms 7-8 in Appendix. We added an explicit reference to those axiom numbers in the revised paper. Thanks for pointing this out.
> - Section 4, line 3: That sentence was confusing, we rephrased it in the revised paper. Our constraint does not encode a multi-label classification problem; it encodes a multi-class classification problem. For example in MNIST, exactly one class from the set {0,1,2,..,9} can be assigned to one picture.
> -Section 4.1: You have the right idea. Semantic loss function is like a penalty term, in a way that it is lowest when the unlabeled point receives exactly one label. For example, in a 3-class classification, the semantic loss value of an output [1,0,0] is smaller than the semantic loss of [0.8,0.1,0.1]. It pushes the unlabeled data in a direction of confidently picking a label.
> - Page 6 and references: These have been corrected, thanks!

---

### Official Review · AnonReviewer1 · 2017-12-01
**Overall the idea is interesting, but the paper does not seem ready for publication.**

**Rating:** 4
**Confidence:** 4

**Review:**

This paper suggest a method for including symbolic knowledge into the learning process. The symbolic knowledge is given as logical constraints which characterize the space of legal solutions.   This knowledge is "injected" into the learning process by augmenting the loss function with a symbolic-loss term that, in addition to the traditional loss, increases the probability of legal states (which also includes incorrect, yet legal, predictions).

Overall the idea is interesting, but the paper does not seem ready for publication.  The  idea of semantic-loss function is appealing and is nicely motivated in the paper, however the practical aspects of how it can be applied are extremely vague and hard to understand.  Specifically, the authors define it over all assignments to the output variables that satisfy the constraints. For any non-trivial prediction problem, this would be at least computationally challenging. The author discuss it briefly mentioning a method by Darwiche-2003, but do not offer much intuition or analysis beyond that.  Their experiments focus on multiclass classification, which implicitly has a "one-vs.-all" constraint, although it's not clear why defining a formal loss function is needed (instead of just taking the argmax of the multiclass net), and even beyond that - why would it result in such significant improvements (when there are a few annotated data points)?

The more interesting case is where the loss needs to decompose over the parts of a structural decision, where symbolic knowledge can help  constrain the output space. This has been addressed in the literature (e.g., [1], [2]) it's not clear why the authors don't compare to these models, or even attempt any meaningful evaluation.


[1] Zhiting Hu, Xuezhe Ma, Zhengzhong Liu, Eduard Hovy, and Eric Xing. Harnessing deep neural
networks with logic rules. ACL, 2016.

[2]Posterior Regularization for Structured Latent Variable Models.  Ganchev et-al 2010.

---

> ### Author Response · Authors · 2017-12-31
> **Response to AnonReviewer1**
>
> Thank you for your valuable feedback.
>
> The proposed semantic loss function is very general, and depending on the constraint you want to enforce, may be computationally challenging. We see this generality as a virtue, and  our goal for this paper was to show the feasibility of semantic loss on a variety of constraints. As you point out, one should avoid evaluating semantic loss by enumerating all assignments to the output variables (this is only feasible for the exactly-one constraint). We avoid this by employing state-of-the-art automated reasoning tools that build efficient circuit representations of the constraint and support efficient weighted model counting (SDD circuits). Unfortunately, here we have to refer to the automated logical reasoning literature for the details of this construction (in particular Darwiche 2011). In the next revision of the paper, we will expand our discussion of this issue. Note however that the slowdown from adding semantic loss to our experiments was negligible, given an initial overhead to build the circuit. We would also like to point out that we are the first to bring these logical circuit techniques to bear on deep learning, which explains why these useful reasoning techniques may not yet be known to the deep learning community.
>
> Taking the argmax of the multiclass net helps to classify instances but does not help to learn from unlabeled data, which is the problem we consider. Just taking the argmax for the unsupervised data does not give us any more information because we do not know if our prediction is correct or not. For example in the 3-class classification case, when the neural network outputs a [0.7, 0.5, 0.4], existing loss functions will still push the label to its correct output [1, 0, 0]. However, in the semi-supervised learning, the unsupervised dataset does not have a label.
>
> Nevertheless our semantic loss function is defined on unsupervised data and captures the inherent constraints of the unsupervised data, that it must have some label. In the revision of the paper we posted, we added more discussion of how semantic loss helps in semi-supervised learning (See Figure 3). It forces the neural network to confidently choose a label for unlabeled data points. In that sense, semantic loss training is like self-training, except that for self-training, once an unlabeled data point is erroneously labeled, it can no longer recover. We also refer to entropy-based regularization for semi-supervised learning as a related idea, which our loss function generalizes to arbitrary constraints (Grandvalet & Bengio, 2005).
>
> We agree that a comparison with related work was missing from the initial submission. We have now added an extensive discussion of related work and how it is conceptually different from semantic loss in the revised paper we posted (see Section 6). We have also included 17 new references.
>
> Specifically for the Hu et al. paper you mention, the key difference with semantic loss is that Hu et al. use fuzzy logic. This has two implications. First, the fuzzy loss function is very sensitive to the syntax of the logical constraint, whereas our loss only depends on the semantics. Second, the logical constraints supported by these fuzzy alternatives are much more simple than the ones we consider. For example, for the Grids experiment in our paper, the constraint is very complex (it doesn’t even have a compact CNF form), and needs to be represented as a logical circuit (see Nishino et al. 2017). We are not aware of a reasonable fuzzy logic encoding. In contrast, other work that uses fuzzy logic to encode constraints works with very simple logical sentences (usually simple implications (X => Y) or Horn clauses). We have also attempted to compare experimentally on the benchmarks of Hu et al. Initial experiments suggested that semantic loss outperforms the loss used by Hu et al. on their evaluation tasks. However, because we found it difficult to exactly reproduce the initialization of Hu et al. and were not able to perform enough tuning, we prefer to not report on that experiment at this time.

---

### Decision · Program_Chairs · 2018-01-29
**ICLR 2018 Conference Acceptance Decision**

**Decision:**

Reject

**Comment:**

This one was really on the fence.  After some additional rounds of discussion post-rebuttal with the reviewers I think the general consensus is that it's a good paper and almost there but not quite ready for acceptance at this time.  A detailed list of issues and concerns below.

PROS:
1. good idea: an additional loss term that enforces semantic constraints on the network output (like exactly 1 output element must be 1).
2. well written generally
3. a nice variety of different experiments

CONS:
1. paper organization.  The authors start with the axioms they would like a semantic loss function to obey, then provide a general definition, then show it does obey the axioms.  The general definition is intractable in a naive implementation.  The authors use boolean circuits to tractably solve the problem but this isn't discussed enough and it's unreasonable to expect readers to just give a pass on it without some more background.  I personally would prefer an organization that presented the motivation (in english) for the loss definition; then the  definition with a description of its pieces and why they are there; then a short discussion of how to implement such a loss in practice using boolean circuits (or if this is too much put it in the appendix); and a pointer to the axiomatization in an appendix.

2. related to 1, I didn't see anything which discussed the training time of this approach.  Given that the semantic loss has to be computed in a more involved way than usual, it's not clear whether it is practical.